# Differentiation Model for Insomnia Disorder and the Respiratory Arousal Threshold Phenotype in Obstructive Sleep Apnea in the Taiwanese Population Based on Oximetry and Anthropometric Features

**DOI:** 10.3390/diagnostics12010050

**Published:** 2021-12-27

**Authors:** Cheng-Yu Tsai, Yi-Chun Kuan, Wei-Han Hsu, Yin-Tzu Lin, Chia-Rung Hsu, Kang Lo, Wen-Hua Hsu, Arnab Majumdar, Yi-Shin Liu, Shin-Mei Hsu, Shu-Chuan Ho, Wun-Hao Cheng, Shang-Yang Lin, Kang-Yun Lee, Dean Wu, Hsin-Chien Lee, Cheng-Jung Wu, Wen-Te Liu

**Affiliations:** 1Department of Civil and Environmental Engineering, Imperial College London, London SW7 2AZ, UK; ct619@ic.ac.uk (C.-Y.T.); a.majumdar@imperial.ac.uk (A.M.); 2Department of Neurology, Shuang Ho Hospital, Taipei Medical University, New Taipei 235041, Taiwan; yckuang2@gmail.com (Y.-C.K.); 19198@s.tmu.edu.tw (C.-R.H.); tingyu02139@gmail.com (D.W.); 3Department of Neurology, School of Medicine, College of Medicine, Taipei Medical University, Taipei 110301, Taiwan; 4Department of Neuropsychology and Cognitive Function, Taipei Neuroscience Institute, Taipei Medical University, Taipei 110301, Taiwan; 5Sleep Center, Shuang Ho Hospital, Taipei Medical University, New Taipei 235041, Taiwan; m141107003@tmu.edu.tw (K.L.); yuki8637@gmail.com (S.-M.H.); 14667@s.tmu.edu.tw (C.-J.W.); 6School of Medical Laboratory Science and Biotechnology, College of Medical Science and Technology, Taipei Medical University, Taipei 110301, Taiwan; b614105010@tmu.edu.tw; 7Department of Medical Imaging and Intervention, Chang Gung Memorial Hospital at Linkou, Taoyuan 333423, Taiwan; linyintzu@cgmh.org.tw; 8Master Program in Thoracic Medicine School of Respiratory Therapy, College of Medicine, Taipei Medical University, Taipei 110301, Taiwan; b117105061@tmu.edu.tw; 9School of Respiratory Therapy, College of Medicine, Taipei Medical University, Taipei 110301, Taiwan; m141109002@tmu.edu.tw (Y.-S.L.); shu-chuan@tmu.edu.tw (S.-C.H.); young19820822@gmail.com (S.-Y.L.); 10Graduate Institute of Medical Sciences, College of Medicine, Taipei Medical University, Taipei 110301, Taiwan; d119106011@tmu.edu.tw; 11Division of Pulmonary Medicine, Department of Internal Medicine, Shuang Ho Hospital, Taipei Medical University, New Taipei 235041, Taiwan; leekangyun@tmu.edu.tw; 12Department of Psychiatry, Shuang Ho Hospital, Taipei Medical University, New Taipei 235041, Taiwan; ellalee@tmu.edu.tw; 13Department of Otolaryngology, Shuang Ho Hospital, Taipei Medical University, New Taipei 235041, Taiwan; 14Professional Master Program in Artificial Intelligence in Medicine, College of Medicine, Taipei Medical University, Taipei 110301, Taiwan

**Keywords:** insomnia disorder, obstructive sleep apnea, in-laboratory polysomnography, respiratory arousal threshold, random forest

## Abstract

Insomnia disorder (ID) and obstructive sleep apnea (OSA) with respiratory arousal threshold (ArTH) phenotypes often coexist in patients, presenting similar symptoms. However, the typical diagnosis examinations (in-laboratory polysomnography (lab-PSG) and other alternatives methods may therefore have limited differentiation capacities. Hence, this study established novel models to assist in the classification of ID and low- and high-ArTH OSA. Participants reporting insomnia as their chief complaint were enrolled. Their sleep parameters and body profile were accessed from the lab-PSG database. Based on the definition of low-ArTH OSA and ID, patients were divided into three groups, namely, the ID, low- and high-ArTH OSA groups. Various machine learning approaches, including logistic regression, k-nearest neighbors, naive Bayes, random forest (RF), and support vector machine, were trained using two types of features (Oximetry model, trained with oximetry parameters only; Combined model, trained with oximetry and anthropometric parameters). In the training stage, RF presented the highest cross-validation accuracy in both models compared with the other approaches. In the testing stage, the RF accuracy was 77.53% and 80.06% for the oximetry and combined models, respectively. The established models can be used to differentiate ID, low- and high-ArTH OSA in the population of Taiwan and those with similar craniofacial features.

## 1. Introduction

Obstructive sleep apnea (OSA) and insomnia are both common sleep disorders that generally coexist in patients [1]. A review in 2019 of 37 related studies documented that the worldwide prevalence of insomnia in patients with OSA ranged from 18% to 42% [2]. A population-based epidemiological study investigated the cultural factors that affect insomnia by comparing the prevalence of insomnia in adolescents in the United States and in Hong Kong, and demonstrated similar estimates of approximately 9.3% [3]. For OSA, various regional ethnicities were analyzed, and the results suggested that Asians and Hispanic/Mexican Americans tended to have an elevated prevalence of OSA compared with Europeans and African Americans [4]. A 2021 review reported that the reduced physical activity and elevated food intake due to the pandemic and consequent lockdowns were associated with an increase in the obese population, which consequently increased the risk of having OSA [5]. Similarly, stress induced by the pandemic has been observed to increase the prevalence of insomnia [6]. The appropriate medical approaches for treating these sleep disorders must be discerned.

Regarding disease characteristics, OSA induces reduced or ceased airflow during sleep resulting from either the complete or partial obstruction in the upper airway. This respiratory event may lead to the occurrence of arousal, altering the sleep stage from that of deep sleep to light sleep or even awakening [7,8]. The respiratory arousal threshold (ArTH) that determines the ventilatory drive trigger may play a substantial role in arousal occurrence [9]. In approximately one-third of patients with OSA, a low-ArTH led to an early arousal response and triggered a ventilatory drive that terminated the respiratory event [10]. A high-ArTH, however, promotes breathing stability through its reduction in undesired repetitive arousal, but consequently induces severe hypoxemia [11,12]. Insomnia disorder (ID), as defined in the criteria of the Diagnostic and Statistical Manual of Mental Disorders, Fifth Edition (DSM-V) [13,14], presents itself as the difficulty in initiating or maintaining sleep and can be associated with the effects of other mental or sleep disorders. However, individuals exhibiting the phenotypes of OSA or ID typically present similar clinical symptoms and describe similar chief complaints, such as daytime sleepiness, inadequate sleep quality, and depression [15]. This may limit the development of disease diagnosis and any treatment strategy [16]. Additionally, an unsuitable sleep disorder treatment can aggravate disease severity. For example, hypnotics, such as benzodiazepines, have been commonly administered to alleviate insomnia [16]; however, such medication must be avoided by patients with OSA who experience insomnia symptoms because such medications may increase the severity of OSA [17,18]. Furthermore, hypnotics must be cautiously used in light of how they reduce muscle tone as a side effect, which may increase the risk of elevating the arousal threshold and desaturation frequency. Continuous positive airway pressure (CPAP) therapy is the first-line treatment for OSA. However, patients with low-ArTH OSA may require co-administration of hypnotics (e.g., trazodone) to increase CPAP adherence given the discomfort of using the device may interrupt continuous sleep, especially for those patients with OSA and low-ArTH who are easily awakened [10]. The inappropriate use of hypnotics by patients with OSA and high-ArTH may aggravate the severity of their OSA. Hence, the accurate classification of ID and the various ArTH phenotypes of OSA is critical.

Overnight in-laboratory polysomnography (lab-PSG) is typically employed to determine the type of sleep disorder. Although this examination is currently the gold standard, concerns have been raised about the numerous factors potentially affecting the validity of lab-PSG results. More specifically, hospital environmental factors and measurement equipment can cause sleep disturbance that induces unusual sleep patterns. One study described the first-night effect, which represents the tendency for individuals to demonstrate an atypical sleep pattern when sleeping in a new environment. This effect was observed in older adults in the United States with ID (mean age: 67.5 years) during their lab-PSG examination [19]. Other research indicates that the sleep parameters obtained through polysomnography (PSG) could be misestimated as a consequence of environmental factors or the first-night effect [20]. Moreover, a related sleep field study concluded that a lab-PSG cannot be used to evaluate insomnia [21]. 

Additional limitations of the lab-PSG include the lack of space in sleep laboratories and the high costs incurred by the requirement of a technician for continuous sleep monitoring [22]. Hence, the waiting lists for undergoing lab-PSG are usually long, after which treatment is initiated. In the United States, the mean waiting time for conducting sleep disorder treatment following lab-PSG is 11.6 months [23]. To address these restrictions, various self-rated questionnaires have been applied as low-cost alternative measurement tools, such as the Insomnia Severity Index (ISI), Pittsburgh Sleep Quality Index (PSQI), and Epworth Sleepiness Scale (ESS). However, because patients with OSA or ID can exhibit similar daytime symptoms, utilizing the ISI with its cutoff point based on individuals with insomnia may lead to the misclassification of patients with OSA [24]. A community-based study enrolled 244 older adults in Taiwan to assess the efficacy of the PSQI in screening for ID [25], with the results revealing that this questionnaire may be an ineffective tool because of its low specificity. For the ESS, although a related study documented elevated scores in patients with insomnia compared with those in the control group, the researchers determined that this increase was not reliably predictive of an insomnia diagnosis [26]. Low-ArTH has been defined as follows: (1) a minimum oxygen saturation measured using pulse oximetry (SpO_2-min_) of less than 82.5%, (2) apnea–hypopnea index (AHI) measurement of less than 30 events/h, and (3) percentage of hypopnea (hypopnea events/hypopnea events + apnea events, (F-hypopnea)) greater than 58.3% [9]. Patients are classified as having low-ArTH if they had two of the aforementioned criteria. However, these clinical sleep parameters are only available through lab-PSG determination, and no relative questionnaire for classifying OSA in relation to ArTH phenotypes is available.

The home sleep test (HST) is yet another alternative portable measurement tool for differentiating between OSA and ID and is often employed to diagnose insomnia or OSA or trace the treatment improvements of these sleep disorders [27]. However, the efficacy of the device in classifying OSA and ID remains unclear. The HST also has several limitations. For example, according to the recommendations of the American Association of Sleep Medicine (AASM) for using HST, this method may only be suitable for populations without comorbid medical conditions, the presence of which could decrease its accuracy [28]. Furthermore, clinical guidelines for employing the HST to evaluate OSA in patients who are overweight (body mass index (BMI) > 40 kg/m^2^) or older adults (aged >65 years) have not been established [29]. Because of these shortcomings in the current measurements for categorizing ID and low- or high-ArTH OSA, a novel method for precise classification is required.

Machine learning has been widely employed in different medical areas, such as in medical analysis aid systems or advancing diagnostic tools [30], and has demonstrated adequate level of accuracy. Therefore, these approaches can potentially be applied to differentiate the aforementioned sleep disorders, which cause similar symptoms in patients. Previous studies have established several machine learning approaches based on multidimensional input features to perform disease classification. A recent review study, for example, reported that one of the machine learning approaches, clustering, was applied to predict the response of patients with depression who had insomnia and were treated with antidepressants [31]. Another recent relevant study developed a neural network based on electrocardiography signals obtained by using a single-lead device for classification of sleep disorders—ID, epilepsy, or rapid eye movement sleep behavior disorder (RBD)—and exhibited favorable performance [32]. For classifying the severity of OSA, several previously developed models, including k-nearest neighbors (kNN), and various types of neural networks based on the time-domain variables extracted from photoplethysmography data were employed, and the reported accuracy of their most effective model was higher than 90% [33]. Another related study established a random forest (RF) model by using electroencephalographic signals and achieved a high mean accuracy (over 90%) for RBD classification [34]. Those researchers suggested using RF because this ensemble approach was effective in establishing and demonstrating acceptable performance and high reliability for some training strategies, such as boosting and bagging. Additionally, RF provided better interpretation of input parameters compared with other supervised learning approaches (e.g., support vector machine (SVM) or kNN). However, these published methods were trained by using open datasets or physiological signals that may not be obtained easily. Moreover, no currently established models that can separate ID and low- or high-ArTH OSA exist.

To develop novel tools with easily-determined features for classifying OSA phenotypes and ID, their potential predictors must be identified. For example, body profiles, such as BMI and neck and waist circumference, are simple to obtain and could serve as prediction features. Obesity is not markedly associated with ID [35,36] but is with OSA [37]. Age and sex also represent suitable indicators for classification. A study exploring the gender difference in insomnia prevalence among 9851 Hong Kong Chinese reported that women had a higher prevalence of insomnia than men [38]. By contrast, a review study analyzing 24 articles that focused on OSA prevalence in Asian populations revealed that this disease was disproportionately more common in men than in women [39]. Moreover, OSA with insomnia symptoms may be more prevalent in older adults because age is a risk factor for the development of both diseases [40,41]. ID is diagnosed independent of any coexisting OSA, and oximetry-related measurements may thus act as alternative indicators for classifying ID and OSA [42]. For instance, the oxygen saturation of lab-PSG measured using pulse oximetry (SpO_2_) and its derived parameter, oxygen desaturation index (ODI-3%), have been applied as surrogates to evaluate OSA severity [43,44]. For the OSA phenotype classification, research has demonstrated that the SpO_2_ level during lab-PSG examination was highly associated with ArTH type classification [9]. Hence, the variables of sex, age, body profiles, and oximetry parameters can be determined without the use of complicated devices or laboratory equipment and may serve as applicable indicators for establishing classification models. 

In this study, we hypothesized that body profiles and oximetry parameters during sleep can be used as potential features for developing classification models for low-ArTH OSA, high-ArTH OSA, and ID. The primary objective of this study was to establish models for two ArTH phenotypes of OSA and ID through the application of easily accessible parameters. These models were established using various machine learning approaches, namely, logistic regression (LR), kNN, naive Bayes (NB), RF, and SVM. We developed these models with clinically feasible methods to perform the accurate differentiation of ID and low- and high-ArTH OSA. Moreover, the sleep parameters of the ID group and low- and high-OSA groups were subjected to mean comparisons to further assess sleep alteration patterns in different sleep disorder groups. 

## 2. Materials and Methods

### 2.1. Study Population

Figure 1 illustrates the data collection procedure of this study. First, we enrolled participants who reported insomnia as their chief complaint and who underwent lab-PSG in the sleep center of Taipei Medical University Shuang Ho Hospital (New Taipei, Taiwan) between January 2015 and August 2020. 

Other inclusion criteria were as follows: (1) patients were aged 18 to 80 years; (2) the total recording time of their lab-PSG was more than 6 h; (3) and their lab-PSG did not apply continuous positive airway pressure or oral devices. Patients who had undergone surgery for OSA or those with sleep disorders other than ID or OSA, such as rapid eye movement (REM) sleep behavior disorder and limb movement disorder, were excluded. The medical record number of eligible patients was collected for further data analysis. Second, the sleep parameters of the lab-PSG report as well as body profiles (records of age, gender, BMI, and neck and waist circumference) were extracted from the sleep center database. Compared with their Caucasian counterparts of similar OSA severity, patients from the Han Chinese population typically have a lower BMI but more restricted craniofacial features [45]. Another study suggested that Han Chinese ethnicity tends to manifest restricted craniofacial bone size and that Caucasians tend to be more overweight at the same level of OSA severity [46]. A related study analyzed the craniofacial features of severity-matched patients with OSA from various ethnicities. Those outcomes documented that smaller craniofacial bone dimensions were observed in those of Japanese ancestry in Brazil, whereas white Brazilians demonstrated a larger volume in the soft tissue of the upper airway [47]. Thus, based on these prior observations, craniofacial features, which affect the upper airway space, may influence the ArTH level and severity of OSA. However, craniofacial features were not employed as indicators in this study because the participants were all of a single ethnicity within the Taiwanese population. Next, deidentification of patients’ personal information was performed, and the participants were divided into three groups, the ID group and the low- and high-ArTH OSA groups [9,14]. Patients with an AHI score lower than 5 or greater than or equal to 5 were classified into the ID and low-ArTH groups, respectively, based on the relevant classification criteria [9].

### 2.2. PSG Results

In this study, either the Embla N7000 (Natus Medical, Pleasanton, CA, USA) or Embla MPR system (Natus Medical, Pleasanton, CA, USA) was used for lab-PSG examination. The sampling rate of the oximetry of these recording devices was set as 25 Hz. In terms of the software, RemLogic™ 3.41 (Embla Systems, Kanata, ON, Canada) was employed for scoring, with all sleep stages, indices (arousal index (ArI), spontaneous arousal index (SpArI), and respiratory arousal index (RArI)), and other measurements (e.g., SpO_2_) scored by a certified sleep technician in accordance with the 2017 AASM clinical guidelines [48]. Following lab-PSG completion, all data were summarized into a report for each patient. In this study, we established models by using easily accessible sleep parameters that can be measured in a home sleeping environment. Because hospital-based sleep parameters, such as the AHI score (measured using nasal pressure and thermistor signals), percentage of each sleep stage, and total sleep time (TST; measured using brainwave signals), cannot be determined without measurement by a sleep technician, these parameters were not employed in model construction. The oximetry data and its related parameters, which were readily obtained without complicated devices and technician support, were applied in this study. Specifically, the mean and minimal values of SpO_2_ (SpO_2-mean_ and SpO_2-min_) as well as the ratio of accumulation time for an SpO_2_ below 90% and total recording time, namely, the index of SpO_2-<90%_ (I-SpO_2-<90%_), were adopted [49]. Next, because the ODI-3% was derived from the TST and may not be accurately determined without lab-PSG, we introduced a surrogate index, the total recording time ODI-3% (ODI-3%_-TRT_), to infer oxygen desaturation severity; the total number of oxygen desaturation events (>3%) was calculated and divided by the total recording time. 

### 2.3. Statistical Analysis

All statistical analyses in this research were conducted using SPSS Version 18.0 (IBM, Chicago, IL, USA). The Shapiro–Wilk test was first used to examine the normality of the obtained continuous variables. Because the data of each variable were not normally distributed (Shapiro–Wilk test: *p* < 0.05), the Kruskal–Wallis H test was applied to examine the differences in parameters among the three groups. Dunn’s test was also employed to provide a pairwise comparison of the three groups. The level of significance for all statistical analyses was set as 0.05.

### 2.4. Prediction Model

To develop models to aid in the clinical classification of ID and two OSA phenotypes (low- or high-ArTH) based on easily accessible parameters, this study applied the machine learning approaches of LR, kNN, NB, RF, and SVM. All of these techniques have been widely used in sleep disorder diagnosis with adequate accuracy—for instance, AHI prediction [50], OSA severity [51], and ID diagnosis [52]. A schematic illustration of the machine learning approaches employed in this study appears in Figure 2. Technical explanations are as follows. First, LR is a statistical method that is commonly employed in estimating probabilities for classification. kNN is a clustering approach with the classification principle that objects in the same category have similar characteristics (close proximity in the dimension of feature). Hence, this method entails setting up the k values for referencing the type of the k number of near neighbors. For the NB model, the classification technique is based on the assumption of variable independence according to Bayes’ theorem, meaning that its classification mechanism relies on the assumption that the presence of a particular feature in a class is not relevant to the presence of any other feature. Concerning RF, this ensemble approach consists of multiple classification and regression trees (CART), and each CART is trained by a subpart of the dataset. When conducting a classification task, each CART in the RF generates their predicted class, and the class that has the highest votes is considered the prediction of RF. Last, SVM provides a decision boundary (line or hyperplane) that separates the input features into classes. 

In terms of the input features, two types of datasets were employed in the training stage, namely, the oximetry-based model (type 1: oximetry parameters only) and combined model (type2: oximetry and body profile parameters). Figure 3 depicts the model establishment procedure. First, the dataset was divided into the training and testing datasets at a ratio of 80:20. To prevent overfitting and evaluate the performance of the established models, the k-fold cross-validation technique (k = 10) was applied in this phase. The setup details of each model are described as follows. The k value of the kNN model was set as 5, which exhibited the highest accuracy among the k values from 1 to 10. With respect to the RF structure, the number of CART was set as 800. This value was determined through out-of-bag sampling estimation, which can be employed to assess the convergence of the error rate [53,54]. Additionally, the bootstrap technique was applied in the training stage of the RF models [55]. This technique randomly samples a subset from the training dataset to decrease training time and prevent overfitting. For SVM, the Gaussian radial basis function was employed as the kernel function. After completion of the training and validation stages, the models with the highest mean accuracy of validation results were used to predict the testing dataset for evaluating model performance and its feature importance. 

### 2.5. Performance Evaluation

The confusion matrices were computed from the testing data predicted using the selected model. These matrices indicated the values of true-positive, true-negative, false-positive, and false-negative results, which were used to determine performance in relation to accuracy, precision, recall, F1-score, and the area under the receiver operating characteristic curve. In regard to feature importance, only the models that demonstrated the highest accuracy in the oximetry and combined models were further investigated for their input feature contribution to prediction. Different machine learning approaches were used corresponding to specific evaluation methods. For instance, permutation importance was applied to evaluate the feature importance for NB, SVM, and kNN [56]. Next, the method of averaging the impurity decrease was used for assessing the variable importance of RF, and coefficient comparison was employed for LR [57,58].

## 3. Results

### 3.1. Demographics of Study Participants

The baseline characteristics of the collected data from eligible participants are listed in Table 1. A total of 404, 624, and 551 patients were assigned to the ID, low-ArTH phenotype OSA, and high-ArTH OSA groups, respectively. The mean age of the ID group was significantly lower than that of the OSA groups (*p* < 0.05). For the anthropometric profiles, among the three groups, the ID group had the lowest significant mean BMI and neck and waist circumference values (*p* < 0.05), whereas the high-ArTH OSA group exhibited the highest significant values (*p* < 0.05). Similarly, the mean values of the three parameters used to define low-ArTH, namely, the AHI, SpO_2-min_, and F-hypopnea, among the three groups were significantly different (*p* < 0.05). Furthermore, the majority of the high-ArTH group was diagnosed as having severe OSA (87.66%). 

### 3.2. Sleep Architecture and Oximetry

The sleep architecture and oximetry of the three groups are summarized in Table 2. In terms of the sleep architecture, the mean sleep latency of the ID group was significantly higher than that of the two OSA groups (*p* < 0.05). The average minutes of wake after sleep onset (WASO) and the mean percentage of the wake stage in the sleep period time (SPT) of the high-ArTH OSA group were significantly higher than those of the other two groups. Conversely, the mean percentage of the REM stage in the SPT of the high-ArTH OSA group was significantly lower compared with the other groups. Regarding the oximetry parameters, the SpO_2-mean_ and ODI-3% of the three groups were significantly different (*p* < 0.05). In terms of the arousal response, the highest mean values in the ArI and RArI were observed in the high-ArTH OSA group compared with those of the low-ArTH OSA and ID groups (*p* < 0.05). Contrarily, for the SpArI, the high-ArTH OSA group had the lowest significant mean value compared with those of the other two groups (*p* < 0.05). Moreover, the patients with ID exhibited the highest significant mean SpArI. 

### 3.3. Model Construction and Performance

Two types of models (oximetry and combined models) were established based on the machine learning approaches of LR, kNN, NB, RF, and SVM. Table 3 presents a comparison of the cross-validation results among the various approaches in relation to the two model types. First, in terms of the oximetry models, the overall accuracy ranged from 77.04% to 79.57%, with the RF model exhibiting the highest value compared with those of the other approaches. Similarly, with respect to the combined models, RF also outperformed the other five approaches in relation to accuracy (80.6%). Therefore, this study adopted the RF models as the selected models to classify the testing set and investigate the feature importance of both model types.

Table 4 presents the classification results obtained using the RF models with the testing set; the derived confusion matrices are illustrated in Figure 4. The overall accuracy of the combined model was relatively superior to that of the oximetry model (80.06% and 77.53%, respectively). For the accuracy in relation to each disease type, the combined model had relatively higher values compared with those of the oximetry model; the precision and recall rate values of the combined model were also relatively higher. Next, because the RF model demonstrated the highest accuracy in both the oximetry and combined models, the method of averaging the impurity decrease was employed for the evaluation of feature importance. In both model types, the ODI-3%_-TRT_ exhibited the highest values. SpO_2_-min and I-SpO_2-<90%_ had similar feature importance and consisted of the second and third highest values; sex had the lowest values for feature importance in the combined model. 

## 4. Discussion

To accurately classify ID and high- and low-ArTH OSA, this study established prediction models by using various machine learning approaches based on oximetry parameters and anthropometric features for those with craniofacial features matching that of the Taiwanese population. The overall accuracy of the models were compared, and the feature importance of selected models was evaluated. Additionally, we compared the parameters of lab-PSG, anthropometric features, and different types of arousal responses among three disease groups. 

In terms of the overall accuracy of the established modes, RF demonstrated the highest accuracy in the oximetry and combined models. Although no evidence has indicated the superiority of RF over other machine learning approaches, these outcomes can be partially explained by several reasons. RF generally uses the bagging and bootstrapping method, which eliminates classification fluctuation and reduces the time until convergence in the training stage [59]. All of these techniques may reduce the variance in the bias of the order, representing an exact classification strategy [60]. In addition, RF, as an ensemble learning model, has the capacity to enhance classification accuracy through calculation of the prediction from each single decision tree classifier, which was trained using the partitioned dataset. This also constituted an effective method against overfitting and reduced the sensitivity caused by certain data characteristics or noise [61]. Taken together, the RF could be used to complement the differential diagnosis of ID and various ArTH phenotypes of OSA. 

Regarding the accuracy of the established models based on the two data types, RF training based on combined features outperformed the RF model trained only with oximetry data. The likely explanation for this outcome is that the anthropometric features provided more dimensional computations when classifying sleep disorders. Although the oximetry parameters were distinct among the patients with ID and high- and low-ArTH OSA [9,62], computation of the anthropometric features may assist in accurate disease classification. Studies have indicated that patients with low-ArTH OSA were more likely to have a non-obesity characteristic, whereas patients diagnosed as having high-ArTH OSA typically exhibited high BMI values, with men composing the majority [63,64]. Similar studies have reported that patients with ID had no significant relation to obesity, with further research required to verify associations [35,65]. These prior observations were consistent with the considerable differences in the body profile measurements of our enrolled groups in this study. Because significant differences in both body profiles and oximetry details among patients with ID and low- or high-ArTH were noted, computation of the oximetry and anthropometric features together may increase the classification accuracy of these three sleep disorder diseases compared to using oximetry data alone.

In terms of the differences in the sleep architecture and arousal response among the three groups, the high-ArTH group presented the highest values for the WASO, ArI, and RArI. These findings may correspond with the high-ArTH group having the most oxygen desaturation events and lowest SpO_2_-mean. The majority of respiratory events were associated with arousal occurrences, which increase dilator muscle activity during the airway obstruction phase [66]. However, this mechanism may alter the sleep stage, which then constitutes brief awakenings that increase the accumulation of WASO. For the patients with ID, the highest SpArI values were observed, which may be attributable to these patients experiencing elevated activity in terms of their high-frequency brainwave signals, resulting in a marked increase in SpArI occurrences [67]. Furthermore, the ID group exhibited the longest sleep onset time; this is in line with the common ID symptom of experiencing difficulty falling asleep [68]. Other research has documented results similar to the current outcomes. For instance, a systematic review outlined the outcomes of several related studies and revealed that patients with ID exhibited more frequent arousal response and increased WASO compared with healthy sleep control groups [69]. Overall, sleep patterns varied among patients with ID and low- and high-ArTH OSA and included different physiological mechanisms and reactions.

Several limitations must be considered in further research. First, the dataset employed in the present study was limited to a single ethnicity (Taiwanese population), and anthropometric parameters rather than craniofacial features were assessed despite the effects of craniofacial factors on OSA symptoms in Asian populations [45]. Therefore, the established models are applicable only to populations with craniofacial features matching those of the Taiwanese population. In clinical practice, PSG serves as the gold standard in determining sleep patterns for the diagnosis of sleep disorders. Nevertheless, every PSG result determination is conducted through manual scoring by different technicians, potentially resulting in PSG scoring variability and affecting the accuracy of the sleep disorder diagnosis [70]. Although the current dataset was derived from one sleep center that regularly provides interscore training to maintain PSG result consistency, scoring variability may still influence the examination results. The first-night effect, which manifests during the first night of PSG from the change in sleeping environment, and its characterizations can alter sleep architecture and physiology, possibly leading to inaccurate PSG outcomes [19]. To minimize PSG outcome bias, future studies could conduct multiple nights of PSG. Another limitation was the lack of background information about the interaction of OSA and ID. These sleep disorders are recognized as multifactorial, and lifestyle factors (e.g., tobacco and alcohol use, menopausal status, and chronic illness) have been associated with both OSA and ID [71]. Next, this retrospective study employed a database that only generally documented patients who had insomnia complaints, instead of detailed reports of their insomnia subphenotypes, such as nocturnal awakenings, short sleep maintenance, or other conditions. Therefore, this limited the present study to obtaining the results of classifying different OSA phenotypes and insomnia. Hence, future studies with data on the diagnosis of specific subphenotypes of insomnia and OSA should determine the possibility of misclassification between these sleep disorders.

Model improvement is another issue that should be addressed in the next step. Models may exhibit more accurate classification outcomes based on comprehensive input features. However, because of the retrospective nature of the study, several factors that were associated with ID and OSA were not obtained. For instance, the interviews for completing various questionnaires, such as the ISI, PSQI, and ESS, were not performed. Additionally, the descriptions of clinical symptoms in medical records were not in a uniform format, which prevented this study from determining the subjective variables relating to clinical symptoms, such as self-reported snoring level, frequency of breathing pauses, and other symptoms. However, these clinical responses may provide additional significant numeric features for classifying ID and OSA. With the training process based on such additional derived data, the established models may be able to demonstrate higher overall accuracy and thus exhibit improved performance. Additionally, it may be worthwhile to investigate the internal associations and the effect on classification of the aforementioned factors. Hence, a dataset with comprehensive dimensional features including information on personal habits, comorbidities, and additional anthropometric parameters must be compiled prior to the development of a novel classification model. Comprehensive measurement that considers objective measurements and subjective parameters, including self-rated sleep pattern, questionnaire outcomes, and other parameters, should be employed. 

## 5. Conclusions

To develop models for differentiating OSA phenotypes and ID with easily accessible parameters, the present study collected a dataset including 1579 Taiwanese patients with insomnia patterns. Two types of models, including one type only applying oximetry parameters and one type applying both oximetry and anthropometric parameters, were established through the use of five machine learning approaches (LR, kNN, NB, RF, and SVM). RF demonstrated the highest accuracy among all the approaches in both types of models, and ODI-3%_-TRT_ had the highest feature importance on classification. The combined-type RF model was superior to that of the oximetry-type RF model. Additionally, various significant differences in anthropometry and sleep parameters among the three disease groups were observed. These established models can be applied to patients with craniofacial features matching those of the Taiwanese population to assist in the classification of ID and low- and high-ArTH OSA and can potentially be used in medical institutions lacking PSG facilities because the models are based on easily obtainable parameters.

## Figures and Tables

**Figure 1 diagnostics-12-00050-f001:**
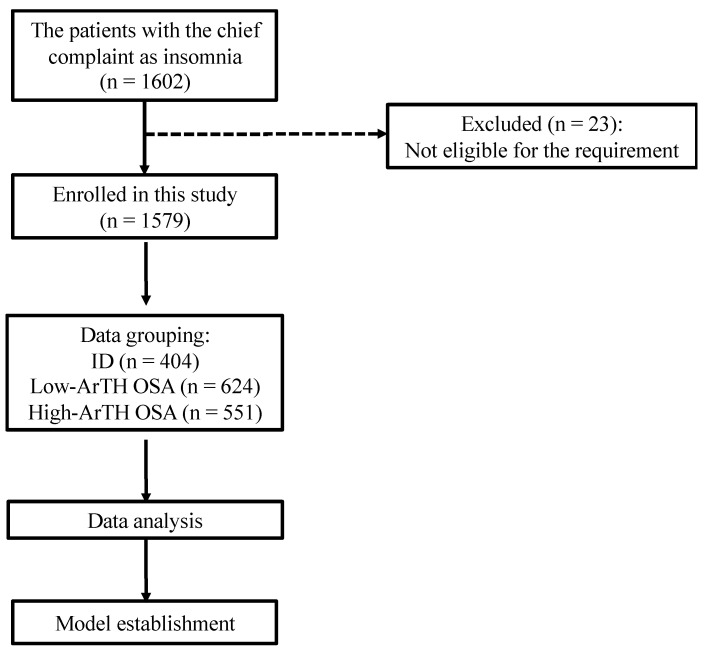
Flowchart of the data collection in the study. The data collection procedure involved included and excluded criteria and eligible data amounts. The data of 1602 patients reporting insomnia as their chief complaint were collected, and 1579 of them were eligible for further analysis. Abbreviation: ID: insomnia disorder; ArTH: respiratory arousal threshold; OSA: obstructive sleep apnea.

**Figure 2 diagnostics-12-00050-f002:**
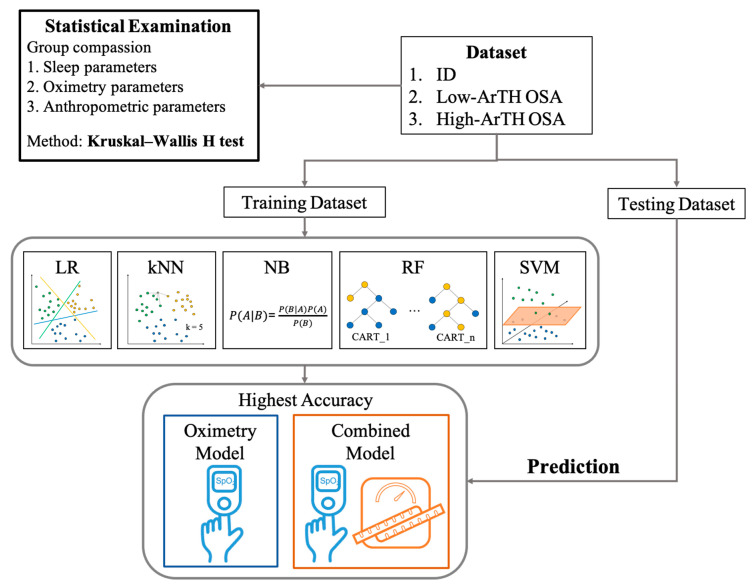
Schematic workflow. All derived data were first subjected to mean comparison for the three disease groups. Next, all data were divided into training and testing datasets. Five matching learning approaches were employed in this study, namely, a logistic regression (LR) model, k-nearest neighbors (kNN), naive Bayes (NB), random forest (RF), and support vector machine (SVM). Two types of models were established by using these five approaches, including the oximetry model, which was only trained using oximetry parameters, and combined models, which were trained by both oximetry and anthropometric parameters. The trained model with the highest accuracy in the training outcomes for both model types was used to predict the testing dataset. Abbreviations: ID: insomnia disorder; ArTH: respiratory arousal threshold; OSA: obstructive sleep apnea.

**Figure 3 diagnostics-12-00050-f003:**
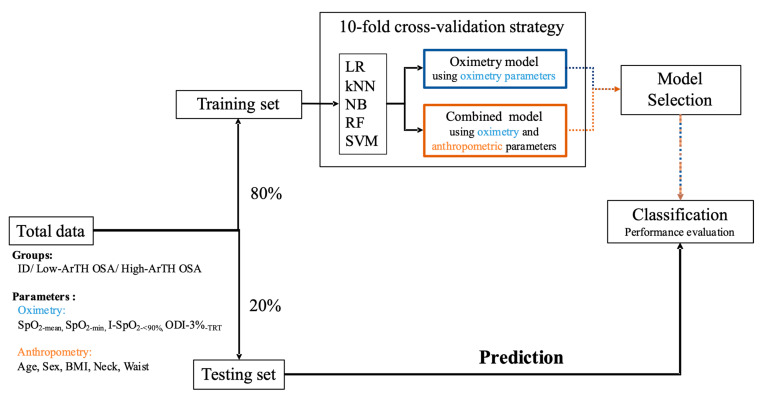
Model establishment process. All data were separated at a ratio of 80:20 to act as the training–validation dataset and testing dataset. The k-fold cross-validation technique (k = 10) was employed in the training–validation stage to separately develop two types of models through five machine learning approaches. Subsequently, the models with the highest accuracy in oximetry typing and combined parameter typing were respectively used to predict the testing dataset for evaluating the overall accuracy and input feature importance. Abbreviations: ID: insomnia disorder; ArTH: respiratory arousal threshold; OSA: obstructive sleep apnea; LR: logistic regression; kNN: k-nearest neighbors; NB: naive Bayes; RF: random forest; SVM: support vector machine; SpO_2-mean_: mean level of peripheral arterial oxygen saturation measured using pulse oximetry; SpO_2-min_: minimum level of peripheral arterial oxygen saturation measured using pulse oximetry; I-SpO_2-<90%_: index for the ratio of the amount of time during which the peripheral arterial oxygen saturation measured using pulse oximetry is lower than 90% to the sleep period time; ODI-3%_-TRT_: total number of oxygen desaturation events (>3%) divided by the total recording time; BMI: body mass index.

**Figure 4 diagnostics-12-00050-f004:**
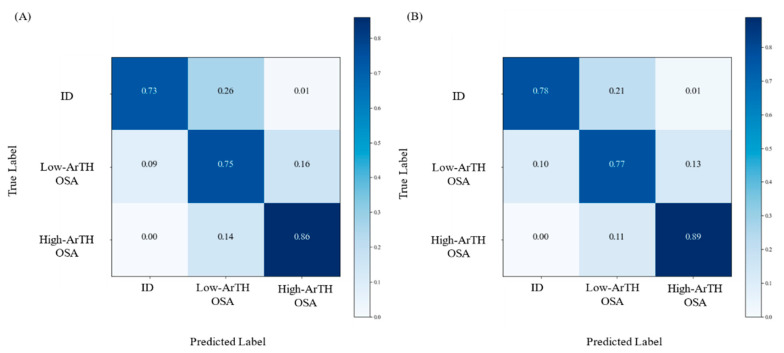
Classification results of the testing set using the selected models (random forest, RF). Confusion matrices indicating the accuracy of the multiclass classification using the selected model (RF). (**A**) The prediction results based on the participants’ oximetry-related parameters (oximetry model); (**B**) The outcomes of the classification model with the anthropometric and oximetry-related parameters (combined model). Abbreviation: ID: insomnia disorder; ArTH: respiratory arousal threshold; OSA: obstructive sleep apnea.

**Table 1 diagnostics-12-00050-t001:** Demographic characteristics of the participants grouped by arousal threshold criteria.

Categorical Variable	ID (*n* = 404)	Low-ArTH OSA (*n* = 624)	High-ArTH OSA (*n* = 551)
Sex (male/female) ^b^	137/267	361/263	422/129
Age (years) ^a^	45.91 ± 13.81 ^#,^^△^	52.42 ± 13.88 ^#^	53.47 ± 13.02 ^△^
BMI (kg/m^2^) ^a^	23.22 ± 4.03 ^#,^^△^	26.06 ± 4.52 ^#,^*	28.08 ± 4.87 *^,^^△^
Neck (cm) ^a^	34.05 ± 3.37 ^#,^^△^	36.74 ± 3.70 ^#,^*	38.81 ± 3.90 *^,^^△^
Waist (cm) ^a^	80.51 ± 10.25 ^#,^^△^	89.69 ± 10.73 ^#,^*	96.13 ± 12.30 *^,^^△^
Low-ArTH criteria ^a^			
AHI (events/h)	2.21 ± 1.48 ^#,^^△^	15.52 ± 7.01 ^#,^*	49.57 ± 21.71 *^,^^△^
SpO_2-min_ (%)	92.44 ± 2.96 ^#,^^△^	86.74 ± 5.26 ^#,^*	78.63 ± 8.30 *^,^^△^
F-hypopnea (%)	45.25 ± 10.94 ^#,^^△^	91.21 ± 11.18 ^#,^*	67.28 ± 29.79 *^,^^△^
OSA severity ^b^			
Mild, *n* (%)	-	334 (53.53%)	30 (5.44%)
Moderate, *n* (%)	-	290 (46.47%)	38 (6.90%)
Severe, *n* (%)	-	-	483 (87.66%)

Abbreviations: ID: insomnia disorder; ArTH: respiratory arousal threshold; OSA: obstructive sleep apnea; AHI: apnea–hypopnea index; SpO_2-min_: minimum level of peripheral arterial oxygen saturation measured using pulse oximetry; F-hypopnea: percentage of respiratory events that were hypopneas. Data are expressed as mean ± standard deviation. ^a^ Differences among groups were assessed using the Kruskal–Wallis H test. ^b^ Differences among groups were assessed using the chi-square test. ^#^ *p*-value was less than 0.05 between the ID and low-ArTH OSA group. **^△^**
*p*-value was less than 0.05 between the ID and high-ArTH OSA group. * *p*-value was less than 0.05 between the low-ArTH OSA and high-ArTH OSA group.

**Table 2 diagnostics-12-00050-t002:** Comparison of the PSG parameter results among the three groups.

Categorical Variable	ID (*n* = 404)	Low-ArTH OSA (*n* = 624)	High-ArTH OSA (*n* = 551)
Sleep architecture			
Sleep onset (min)	30.42 ± 36.79 ^#,^^△^	26.72 ± 31.6 ^#^	24.93 ± 34.11 ^△^
WASO (min)	59.67 ± 55.54 ^#,^^△^	62.81 ± 49.12 ^#,^*^,^	72.11 ± 55.46 *^,^^△^
Wake (% of SPT)	17.75 ± 16.35 ^△^	18.63 ± 14.58 *	21.65 ± 17.21 *^,^^△^
NREM (% of SPT)	71.87 ± 14.37	70.97 ± 12.85	69.84 ± 15.39
REM (% of SPT)	10.38 ± 6.51 ^△^	10.41 ± 6.43 *	8.52 ± 5.93 *^,^^△^
Oximetry parameters			
SpO_2-mean_ (%)	97.12 ± 1.12 ^#,^^△^	96.01 ± 1.35 ^#,^*	94.61 ± 2.13 *^,^^△^
ODI-3% (events/h)	1.82 ± 1.54 ^#,^^△^	12.97 ± 7.61 ^#,^*	46.99 ± 22.28 *^,^^△^
Arousal parameters (events/h)			
ArI	13.67 ± 9.07 ^#,^^△^	17.34 ± 9.12 ^#,^*	30.89 ± 18.36 *^,^^△^
SpArI	11.34 ± 7.68 ^#,^^△^	9.49 ± 6.41 ^#,^*	6.80 ± 6.50 *^,^^△^
RArI	0.62 ± 0.97 ^#,^^△^	5.99 ± 4.35 ^#,^*	22.22 ± 16.62 *^,^^△^
SnArI	0.13 ± 0.65 ^#,^^△^	0.32 ± 1.02 ^#^	0.40 ± 1.40 ^△^
LMArI	1.40 ± 3.11	1.38 ± 2.06	1.35 ± 2.13

Abbreviations: ID: insomnia disorder; ArTH: respiratory arousal threshold; OSA: obstructive sleep apnea; WASO: wake after sleep onset; SPT: sleep period of time; NREM: nonrapid eye movement; REM: rapid eye movement; SpO_2-mean_: mean level of peripheral arterial oxygen saturation measured using pulse oximetry; ODI-3% (events/h): greater than or equal to the 3% oxygen desaturation index; ArI: arousal index; SpArI: spontaneous arousal index; RArI: respiratory arousal index; SnArI: snoring arousal index; LMArI: limb movement-related arousal index. Data are expressed as mean ± standard deviation. Differences among groups were assessed using the Kruskal–Wallis H test. ^#^ *p*-value was less than 0.05 between the ID and low-ArTH OSA group. **^△^**
*p*-value was less than 0.05 between the ID and high-ArTH OSA group. * *p*-value was less than 0.05 between the low-ArTH OSA and high-ArTH OSA group.

**Table 3 diagnostics-12-00050-t003:** Comparison of the cross-validation results of the two model types established using various machine learning approaches.

Categorical Variable	LR	kNN	NB	RF	SVM
Training set (*n*)	ID: 332; Low-ArTH OSA: 416; High-ArTH OSA:515				
Oximetry model					
Accuracy (%)	77.28	77.67	77.20	79.57	77.04
Precision (%)	77.25	78.17	79.03	80.38	77.88
Recall (%)	78.79	78.60	78.71	80.65	77.07
F1-score (%)	77.59	78.16	77.79	80.11	77.31
AUC (%)	91.82	89.60	91.79	92.52	90.99
Combined model					
Accuracy (%)	79.65	71.65	77.75	80.60	78.70
Precision (%)	79.57	72.89	79.35	81.88	79.73
Recall (%)	81.05	71.73	78.94	81.48	78.66
F1-score (%)	79.97	72.05	78.34	81.19	78.98
AUC (%)	93.17	86.96	91.16	93.40	92.01

Abbreviations: LR: logistic regression; kNN, k-nearest neighbors; NB: naive Bayes; RF: random forest; SVM: support vector machine; ID: insomnia disorder; ArTH: respiratory arousal threshold; OSA: obstructive sleep apnea; AUC: area under the curve.

**Table 4 diagnostics-12-00050-t004:** Classification performance and feature importance of the selected models (random forest) in the prediction of the testing set.

Categorical Variable	Oximetry Model	Combined Model
Testing set (*n*)	ID: 72; Low-ArTH OSA:109; High-ArTH OSA:135	
Accuracy (%)	77.53	80.06
Precision (%)	78.72	80.17
Recall (%)	78.17	81.24
F1-score (%)	78.14	80.41
AUC (%)	92.24	93.61
Feature importance (%)		
SpO_2-mean_	5.43	6.46
SpO_2__-min_	14.89	14.69
I-SpO_2-<90%_	15.10	13.40
ODI-3%_-TRT_	64.57	49.50
Age	-	3.25
BMI	-	4.06
Neck	-	3.29
Waist	-	4.51
Sex	-	0.84

Abbreviations: ID: insomnia disorder; ArTH: respiratory arousal threshold; OSA: obstructive sleep apnea; AUC: area under the curve; SpO_2-mean_: mean level of peripheral arterial oxygen saturation measured using pulse oximetry; SpO_2-min_: minimum level of peripheral arterial oxygen saturation measured using pulse oximetry; I-SpO_2-<90%_: the ratio of the amount of time during which the peripheral arterial oxygen saturation measured using pulse oximetry is lower than 90% to the sleep period time; ODI-3%_-TRT_: total number of oxygen desaturation events (>3%) divided by the total recording time; BMI: body mass index.

## Data Availability

All the data of this study were collected from the Sleep Center of Shuang Ho Hospital, New Taipei City, Taiwan, between January 2015 and August 2020. Since there is personal information within the dataset, it is not available in the supplement file. Please get in touch with the correspondence to require the dataset or relevant information if needed.

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
