# Peer review of "Differentiation Model for Insomnia Disorder and the Respiratory Arousal Threshold Phenotype in Obstructive Sleep Apnea in the Taiwanese Population Based on Oximetry and Anthropometric Features"

_diagnostics, 2021, doi:10.3390/diagnostics12010050_

Round 1

Reviewer 1 Report

Thank you for this very nice work. I don’t have a lot of comments because the paper is well written and organized.

Here are my questions / comments:

  • The introduction is very clear and well organized, just one question : what are the difference of outcome and/or treatments (or risks) between the low ArTH and high ArTH OSA? Few background about that would be interesting.
  • Study population : Since it’s a major criteria for patient selection, it would be great to be more specific on the definition for “insomnia complaints” : high sleep latency ? early awakenings ? Nocturnal awakenings ? Duration of these sleep difficulties ? It would be great to have a description of theses subphenotypes of insomnia to describe the ID subgroup. And if the different kind of sleep difficulties are well represented in the ID  group (only IF), I have a further interrogation for the result part : when there are misclassifation by the system between OSA and ID, does it happen more for some kind of insomnia ?

  • Methods : PSG results (lines 189-209) : It would be great to have the sampling frequency of the saturation record.

  • In the discussion part, you talk about the idea to implement results from some questionnaires to enhance the machine learning efficacy and specificity, that’s really the next step and you don’t have it for your sample, OK. But maybe could you have some clinical data from medical interview that might be usefull i.e. if patients report snoring or breathing pauses during sleep ? 

Thank you by advance for your answers.

Author Response

We have carefully revised our manuscript per your valuable suggestions.

Reviewer 2 Report

This topic and approach are very promising and shall be considered for publication, however I have few remarks that need to be addressed before publication:

  1. line 52 - references on Prevalence of obstructive sleep apnea (OSA) and insomnia shall be more current (2018/2015/2019/2010 [1-4] not current enough, the global pandemics might have affected this)
  2. authors shall consider comparing the OSA and Insomnia prevalence variance according to geographical or cultural regions
  3. in general, the literature references in the introduction are not up-to date, at least few references add from year 2020-2021
  4. please add table of abbreviations
  5. line 66 "DSM-IV criteria" add reference + abbreviation in full text?
  6. Introduction shall at least briefly describe the mentioned various machine learning approaches, with literature references. There are many studies using ML for OSA and ID identification. 
  7. please consider adding more references on comparison of craniofacial features between Taiwan population and Caucasian type, as well as on BMI comparison. Reference [38] is not sufficient
  8. Figure 2. Model establishment process is confusing, please consider to reorganize this scheme, including the text placements
  9. line 222 add more references than [42]
  10. more elaborate AI techniques employed, *lines 230-240
  11. Authors repeat that different machine learning approaches were used, however these need to be explained in more detail.
  12. Discuss how the approach in this retrospective study could be improved and elaborate more on RF in the introduction
  13. Reduce the length of Conclusion chapter if possible.
  14. Add a schematic figure about AI approaches used and the workflow
  15.  Unless significantly increasing referencing on ML approach in this article, consider changing the article title more appropriately.

In general, this article is interesting and shall be clinically useful. Authors shall consider to better explain the AI methods employed in the evaluation including the data preparation and training. 

The number of coauthors is extensive, revise if all of the coauthors contributed significantly to this research. Reduction recommended.

Author Response

Thanks for your precious comments.

We have carefully reviewed your manuscript according to your valuable suggestions. Please see the attachment.

Round 2

Reviewer 1 Report

Thank you for revising the manuscript according my suggestions.

Have a nice day.

Author Response

Dear respected reviewer, 

We did appreciate your effort and valuable feedback on this manuscript.

Sincerely,

Reviewer 2 Report

Dear authors,

it was pleasure to read your revised manuscript. It will be a valuable contribution in this topic. You have revised it properly. I recommend its publication.

The last remark is possible placing this article in the other medical context,  with reference that various machine learning approaches are currently successfully applied in medical fields advancing diagnostic methods. Ideally at the beginning or the end of the introductory chapter you can mention recently published article about Use of advanced artificial intelligence in fields from Clinical anatomy and Anthropology to Forensic Medicine.

Kind regards

Author Response

Dear reviewer,

We have carefully reviewed your manuscript according to your suggestion. Please see the attachment.
